# Sex-Related Differences in Immunotherapy Toxicities: Insights into Dimorphic Responses

**DOI:** 10.3390/cancers17071054

**Published:** 2025-03-21

**Authors:** Jacopo Canzian, Fabio Conforti, Flavia Jacobs, Chiara Benvenuti, Mariangela Gaudio, Riccardo Gerosa, Rita De Sanctis, Alberto Zambelli

**Affiliations:** 1Department of Biomedical Sciences, Humanitas University, Pieve Emanuele, 20072 Milan, Italy; jacopo.canzian@humanitas.it (J.C.); chiara.benvenuti@humanitas.it (C.B.); mariangela.gaudio@humanitas.it (M.G.); riccardo.gerosa@humanitas.it (R.G.); 2Humanitas Cancer Center, IRCCS Humanitas Research Hospital, Rozzano, 20089 Milan, Italy; 3Division of Medical Oncology, Humanitas Gavazzeni, 24125 Bergamo, Italy; fabio.conforti@gavazzeni.it; 4Division of Medical Oncology, Fondazione IRCCS Istituto Nazionale dei Tumori, 20133 Milan, Italy; flavia.jacobs@istitutotumori.mi.it; 5Oncology Unit, ASST Papa Giovanni XXIII Hospital, 24127 Bergamo, Italy; alberto.zambelli@unimib.it; 6Department of Medicine and Surgery, University of Milano-Bicocca, 20126 Milan, Italy

**Keywords:** sex dimorphism, immune checkpoint inhibitors, immune-related adverse events

## Abstract

The gender gap in oncology is gaining increasing attention, affecting both the efficacy and toxicity of treatments, including immunotherapy. In this review, we analyze the key biological and clinical mechanisms underlying sex differences in response to immunotherapeutic drugs, with a particular focus on immune checkpoint inhibitors. We highlight the importance of considering sex as a fundamental variable in clinical practice and trial design, emphasizing the need for a more refined approach toward precision oncology.

## 1. Introduction

In oncology, patient outcomes depend not only on the use of effective cancer therapies but also on the ability to recognize and manage treatment-related toxicities. Sex- and gender-based differences in treatment toxicity remain underexplored in clinical practice.

“Sex” refers to biological attributes such as chromosomal patterns (e.g., XX for females and XY for males), reproductive anatomy, hormone levels, and secondary sexual characteristics. In contrast, “gender” is a broader and more complex concept that encompasses roles, behaviors, and identities shaped not only by biological factors but also by social, cultural, and psychological influences.

Sex-based differences in oncology span epidemiology, tumor biology, and treatment toxicity [1]. The distinct susceptibility of men and women to cancer therapies has long been recognized, significantly impacting quality of life, treatment response, and, ultimately, therapeutic success. For instance, chemotherapy has been associated with greater toxicity in women, including a higher risk of acute hematologic and non-hematologic adverse events [2]. The potential biological factors underlying these sex-specific toxicity profiles are complex and involve the differential effects of genetic, epigenetic, and hormonal influences on drug pharmacokinetics and pharmacodynamics [2].

A key challenge in addressing sex differences in drug toxicity is the historical underrepresentation of women in clinical trials, largely due to concerns about reproductive health and potential hormonal variability affecting study outcomes [3]. As a result, current clinical guidelines on dosing and toxicity are largely based on data from predominantly male populations, creating a significant gap in understanding the sex-specific effects of cancer therapies. Conversely, the frequent underestimation of toxicities in men—whether due to underreporting or differences in clinical presentation—may result in inadequate monitoring and management.

Advancing our understanding of sex- and gender-specific differences in drug effects and toxicity, alongside improving their reporting, is essential for creating personalized treatments and optimizing oncology outcomes.

## 2. Methods

We conducted a comprehensive review of the published English literature across PubMed/MEDLINE, Embase, and Cochrane Library, including all publications until 31 January 2025. We used the following search terms: ((gender OR sex OR “sex differences” OR “gender differences” OR female OR male OR women OR men) AND (immunotherapy OR “immune checkpoint inhibitor” OR “checkpoint blockade” OR “PD-1” OR “PD-L1” OR “CTLA-4”)) AND (efficacy OR effectiveness OR “adverse events” OR “immune-related adverse events” OR “irAE” OR toxicity OR “immune toxicity”). We further augmented our search by manually reviewing reference lists of relevant clinical and preclinical studies and review articles. Disagreements were resolved through discussion among the authors.

## 3. Biological Factors Underpinning Gender-Related Differences in Drug Toxicities: The Role of Sex Hormones and Pharmacokinetics

Sex-specific differences in pharmacokinetics and pharmacodynamics can lead to variations in the absorption, distribution, metabolism, and elimination of anticancer drugs [4]. Consequently, sex may influence the active drug dose and patient exposure, leading to variations in toxicity profiles for chemotherapeutic agents, as well as for targeted therapies and immunotherapies [5].

Among patients receiving a standard drug dose, females generally exhibit higher plasma drug concentrations and prolonged elimination times, potentially increasing toxicities and suggesting that women may be routinely overmedicated [6].

One of the parameters commonly used in dose calculations is the body surface area (BSA). However, the BSA does not account for differences in body fat percentage, which is typically higher in females due to genetic and hormonal factors, potentially leading to different drug distribution patterns [7].

Sex influences anticancer drug metabolism through variations in the expression and activity of drug-metabolizing enzymes such as CYP3A4, CYP2D6, and CYP2C19. For instance, CYP3A4 activity is approximately 30% higher in women, whereas men exhibit greater CYP1A2 activity, affecting the metabolism of multiple drugs [8]. Furthermore, sex-based differences in the expression of drug transporters, such as P-glycoprotein (P-gp), which regulates drug absorption and excretion, contribute to variability in pharmacokinetics and toxicity profiles [9].

Sex also influences renal excretion, and renal clearance (C_r_) is often estimated using the Cockcroft–Gault formula, which includes sex as a variable. Men typically exhibit a higher glomerular filtration rate (GFR) and renal plasma flow than women, leading to the faster elimination of several anticancer drugs such as 5-fluorouracil (5-FU) [10].

Finally, sex differences in pharmacodynamics have been observed, likely due to variations in drug–receptor affinity, receptor density, and signal transduction pathways [11]. However, few studies have evaluated whether sex can influence the tissue-specific expression of anticancer targets, and further efforts are needed to establish the presence of any pharmacodynamic sex differences that may affect the toxicity profiles of oncological drugs.

## 4. Sex Differences in Immunity and Antitumor Response

### 4.1. Sex Differences and Immunity

Sex dimorphism in the immune system is a complex phenomenon, encompassing susceptibility to cancer, infections, autoimmune diseases (AIDs), and responses to vaccines [12]. These differences are shaped by an interplay of genetic, hormonal, and environmental factors, which modulate immune responses in males and females in intricate ways throughout their lifespans.

The immune system is influenced by the karyotype, with females (XX) and males (XY) exhibiting distinct immunogenetic profiles. The X chromosome encodes several immune-related genes, including toll-like receptor 7 (TLR7) and toll-like receptor 8 (TLR8), which are essential for pathogen recognition and the activation of innate immune responses [13]. Women, possessing two X chromosomes, benefit from mechanisms such as gene dosage compensation and X-inactivation skewing, which contribute to a more diverse and robust immune response. In contrast, men, with only one X chromosome, are more vulnerable to deleterious mutations in immune-related genes, as they lack a second X chromosome to compensate [14]. Genetic disorders such as Turner syndrome (X0) are therefore associated with reduced activation of T-cells and B-cells and lower levels of IgM and IgG compared to XX females. Conversely, Klinefelter syndrome (XXY) correlates with higher lymphocyte count and increased immunoglobulin (Ig) levels compared to XY males [15].

The complexity of sex dimorphism in the immune system arises from both the differential segregation of immune-related genes and the regulatory effects of sex hormones, which are key regulators of innate and adaptive immune responses. Preclinical studies support the role of hormonal influence, demonstrating that immune cell proliferation can be either enhanced or suppressed in response to male and female sex hormones, independent of sex-linked genetic factors [16]. In women, estrogen (ER) and progesterone (PgR) exert different immunomodulatory effects, varying across life stages such as puberty, menstrual cycles, pregnancy, and menopause. These hormonal fluctuations influence immune function and disease susceptibility over time [12].

ER and PgR receptors are expressed in various cells of the lymphoid tissue, including lymphocytes, macrophages, and dendritic cells (DCs) [17,18]. PgR generally exerts anti-inflammatory effects, whereas ER enhances both innate and adaptive immune responses [12]. For instance, 17-β-estradiol (E2) upregulates pattern recognition receptors such as TLR7, leading to increased interferon-alpha (IFN-α) levels, pro-inflammatory cytokines production, and antigen presentation to naïve T cells [19]. Although this enhanced immune response may lead to the faster and more effective clearance of pathogens, it may also result in an exaggerated inflammatory response.

Sex dimorphism in the adaptive immune system affects cell-mediated immunity, immune memory, and self-recognition. Women typically exhibit higher CD4+ T cell counts and an increased CD4+/CD8+ ratio, leading to greater activation of T-cell-related immune pathways [20]. Females also exhibit higher circulating B-cell levels than males [21].

ER further enhances B-cell activity by increasing IgM and IgG production, promoting hypermutation, and facilitating class switch recombination, contributing to stronger vaccination responses in women [22].

Although aging leads to immunosenescence in both sexes, its effects differ between men and women. Older men experience a greater decline in T- and B-cell populations, resulting in a weaker immune response [23].

Genetic and hormonal factors also shape central and peripheral immunotolerance, which is crucial for recognizing self-antigens and preventing autoimmunity. Women are more susceptible to AIDs, accounting for approximately 80% of cases [24]. Thymic negative selection of autoreactive naïve T cells by the autoimmune regulator (AIRE) is crucial for establishing central immune tolerance and has been shown to be sex hormone-dependent and less active in females [25].

E2 further disrupts peripheral immune tolerance by enhancing B-cell-mediated autoantibody production and modulating regulatory T cells (T_reg_) [26]. Further complexity arises from the biphasic influence of E2 concentrations, particularly on T-cell responses. For instance, pregnancy increases the risk of systemic lupus erythematosus (SLE) flares due to heightened Th2 responses, whereas Th1-dominant diseases such as rheumatoid arthritis (RA) and multiple sclerosis (MS) often improve during pregnancy but relapse postpartum [27]. Conversely, testosterone exerts immunosuppressive effects by downregulating TLR expression on macrophages and reducing the production of key pro-inflammatory cytokines, such as tumor necrosis factor (TNF) and interleukin 6 (IL-6), thereby dampening immune responses [28]. Testosterone also modulates the adaptive immune system by promoting T_reg_ development and suppressing antibody responses [29]. Gonadotropin-releasing hormone (GnRH) antagonists decrease T_reg_ cell counts and elevate natural killer (NK) cell levels in the peripheral blood of males [30]. Similarly, castrated male mice show elevated CD4+ and CD8+ T cells, macrophages, and antigen-specific CD8+ T cells compared to gonadally intact mice [31]. This hormonal environment may make men more susceptible to infections and less responsive to vaccines compared to women. However, it also protects against the development of autoantibodies and, consequently, autoimmune diseases.

### 4.2. Sex Differences and Antitumor Response

A significant gap remains in our understanding of how the innate and adaptive immune systems interact with cancer from a sex-based perspective within the tumor microenvironment (TME). The TME is a dynamic niche composed of tumor and immune cells, often presenting distinct immune phenotypes classified as “immune desert”, “immune-excluded”, or “inflamed” tumors, which vary by histology and influence treatment responses [32].

The heterogeneity in anticancer immune responses between males and females may partially result from the effects of sex hormones on immune cells within the TME. However, most of these differences have been studied in mouse models, and their direct applicability to human cells, particularly in cancer patients, remains uncertain.

Both the innate and adaptive immune systems contribute to the intra-tumoral response, with hormonal factors potentially driving sex dimorphism from the earliest stages of tumor antigen presentation. For instance, E2 enhances DC development, increasing their quantity and functionality, which promotes T-cell trafficking, infiltration, and tumor cell recognition within the TME [33]. In contrast, in males, DCs are polarized toward a tolerogenic phenotype due to androgen-mediated mechanisms, resulting in impaired neoantigen presentation [34]. Additionally, men experience poorer immune infiltration, with reduced T-cell receptor (TCR) repertoire clonality, lower expression levels of class I and II Human Leukocyte Antigen (HLA) molecules, and a higher frequency of HLA class I loss of heterozygosity events [35]. Although T-cell activity may be impaired in both male and female TMEs, the underlying mechanisms differ. In females, the TME is marked by T-cell dysfunction despite high cytotoxic T lymphocyte (CTL) infiltration, whereas in males, it exhibits T-cell exclusion, low CTL levels, and a predominantly Th2-skewed immune response [36]. To evade the initially robust antitumor immune response in females, tumors may gradually develop immune evasion mechanisms, resulting in a higher abundance of immunosuppressive cells including macrophages, cancer-associated fibroblasts (CAFs), myeloid-derived suppressor cells (MDSCs) and T_regs_. Notably, T_regs_ tend to be more activated in females, further contributing to an immunosuppressive tumor microenvironment [36].

Despite a “hot” TME, women exhibit higher expression of immune checkpoint molecules, including PD-L1, TIM3, VISTA, and TIGIT, suggesting complex immune evasion mechanisms [36].

Age is an additional factor that modulates the antitumor immune response. In younger female patients, the TME undergoes greater immunoediting, leading to tumors that are less effectively presented to T cells, thereby reducing immune visibility. Consequently, younger females may develop tumors with enhanced immune evasion, potentially limiting the efficacy of immune checkpoint blockade (ICB) [37].

A graphical review of sex-related differences in the antitumor immune response is shown in Figure 1.

## 5. Sex-Based Differences in Efficacy and Immune-Related Toxicities of Immune Checkpoint Inhibitors: Evidence and Insights

While sex differences in chemotherapy susceptibility are well established, research on sex-based disparities in immunotherapy responses remains limited. Over the past decade, immune checkpoint inhibitors (ICIs) have revolutionized treatment options for multiple malignancies, including melanoma, head and neck squamous cell carcinoma (HNSC), non-small cell lung cancer (NSCLC), and renal cell carcinoma (RCC). These agents include anti-programmed death-1 (anti-PD-1) drugs like pembrolizumab and nivolumab; anti-programmed death-1 ligand (PD-L1) drugs such as atezolizumab, avelumab, and durvalumab; anti-cytotoxic T-lymphocyte–associated antigen 4 (CTLA-4) drugs like Ipilimumab and tremelimumab; and anti-lymphocyte activation gene-3 (LAG-3) drugs such as relatlimab [38,39,40,41].

Recent advances in understanding the complex interplay between antitumor immunity and the TME have heightened interest in biomarkers predicting response to ICIs [42]. However, the sex-specific modulation of immune responses induced by ICIs within the TME remains largely underexplored. By engaging the immune system, ICIs drive both the durable responses observed with immunotherapy and its related toxicities. Indeed, alongside their remarkable clinical benefits, ICIs have also been shown to present a unique spectrum of immune-related adverse events (irAEs) which can be severe, potentially leading to treatment discontinuation and impacting patient outcomes [43]. Given the strong influence of genetic and hormonal factors on immune function, assessing sex-based differences in irAE incidence is essential for optimizing the safety and efficacy of immunotherapy.

### 5.1. Sex Differences and ICIs Efficacy

Women typically exhibit greater immune infiltration and antitumor gene expression, which may result in better outcomes across different tumor types. However, this heightened immune surveillance also enhances immunoediting, thereby reducing tumor immunogenicity and potentially limiting the efficacy of ICIs [36].

A systematic review and meta-analysis of over 11,000 patients from randomized controlled trials (RCTs) identified significant sex-based differences in ICI efficacy. Men experienced greater survival benefits from ICI monotherapy, regardless of the cancer type, treatment line, or ICI class (e.g., anti-PD-1 or anti-CTLA-4) [44]. This may be explained by the more excluded yet less suppressive immune profile of the male TME, making them more responsive to ICIs, which reinvigorate T-cell activity [45].

The role of sex in immunotherapy response was further highlighted in a large-scale analysis by Litchfield et al., which identified sex as a significant predictor of ICI efficacy across various tumor types, independent of molecular factors such as tumor molecular burden (TMB), tumor heterogeneity, an abundance of tumor-infiltrating lymphocytes (TILs), and genetic and epigenetic alterations [46].

An analysis of eight RCTs in advanced NSCLC found that men had significantly greater survival with anti-PD-1, anti-PD-L1, or anti-CTLA-4 monotherapy, whereas females benefited more from combined anti-PD-(L)1 and chemotherapy. These findings align with evidence of sex-based differences in anticancer immune responses and immune evasion mechanisms in NSCLC [47,48]. Chemotherapy likely enhances tumor immunogenicity by inducing cell death and improving antigen presentation, thereby heightening immune response in women and overcoming the limitations of ICI monotherapy. Conversely, Wallis et al. found no sex-based differences in ICI efficacy, possibly due to the inclusion of four RCTs that tested anti-PD-1/PD-L1 combined with chemotherapy–trials not considered in the Conforti meta-analysis. All four of these trials demonstrated significant sex-based heterogeneity in favor of women, balancing the male-favored positive effects of ICI monotherapy versus control therapy [49].

These conflicting results may also arise from the predominant inclusion of highly immunogenic tumors, such as melanoma and NSCLC, reflecting the high number of clinical trials investigating ICB in these cancer types. However, melanoma and NSCLC exhibit distinct sex-based immune dimorphism, which could lead to varied responses to ICIs. Additionally, comparing different types of ICIs (anti-PD-1, anti-PD-L1, and anti-CTLA-4) is challenging as their antitumor activity may be differentially affected by sex-based immune differences. This variability increases data heterogeneity, making it difficult to draw definitive conclusions [50,51].

Interestingly, Jang et al. demonstrated that in older patients with metastatic melanoma, women benefited less from dual checkpoint blockade therapy compared to men [52]. The study suggests that age-related changes in the ER signaling pathway may also contribute to the differences in ICI treatment response observed between women and men undergoing ICI combination therapy. Additionally, older patients exhibit improved responses to anti-PD-1 therapy due to reduced T_reg_ infiltration and higher CD8+ T-cell counts, emphasizing that both age and sex shape immunogenic phenotypes and influence immunotherapy outcomes across patient populations [53].

Finally, Conforti et al. demonstrated that while higher TMB generally correlates with improved outcomes in patients receiving anti-PD-1/PD-L1 inhibitors, this relationship differs by sex. In males, the survival benefit is observed only at very high TMB levels, whereas in females, a linear association exists between TMB and survival, suggesting a more efficient immune response and enhanced tumor recognition [36]. Furthermore, in NSCLC patients, anti-PD-1/PD-L1 monotherapy was highly effective in men but not in women, even in tumors with high PD-L1 expression. This suggests that PD-L1 expression may have a sex-specific predictive value, reflecting a more immunosuppressive TME and less responsiveness to ICI monotherapy in female patients [48].

### 5.2. Sex Difference and irAE Incidence

The broad spectrum of irAEs can affect nearly any organ, including the skin, gastrointestinal tract, liver, lungs, and endocrine system [54]. Their type, intensity, and timing depend on the type of ICI used and whether it is administered as a monotherapy or in combination, potentially leading to discontinuation in up to 40% of patients [55,56]. However, predicting the onset, affected organs, or irAE severity remains unreliable, making patient management challenging [57].

IrAEs develop through multiple mechanisms, including aberrant T-cell activation, autoantibody production, molecular mimicry, and the reactivation of pre-existing autoimmunity [54,58]. The most comprehensive analysis on this topic was conducted by Unger et al. from over 200 phase II-III clinical trials, reporting the incidence of both symptomatic and objective adverse events across various treatment modalities. Notably, the study found that women had a 34% higher risk of severe (Grade 3 or higher) adverse events compared to men, with the greatest disparity observed with immunotherapy. Women receiving immunotherapy were particularly vulnerable to symptomatic irAEs, with a 66% increased risk compared to men (OR = 1.66; 95% CI, 1.37 to 2.01; *p* <.001). Women also exhibited a higher incidence of objective irAEs, particularly hematologic, endocrine, gastrointestinal, and cardiac toxicities. Interestingly, objective renal/genitourinary toxicity was more common in men, regardless of the treatment modality [59]. Consistently, two other studies based on FDA Adverse Event Reporting System (FAERS) data demonstrated that males have a higher risk of immune-related renal toxicity, particularly with anti-PD-1 treatments, suggesting a potential male sex bias [60,61].

Since the study by Unger et al. reported system-specific but not organ-specific irAEs, further studies using FAERS data have explored sex-based differences in specific immune-related toxicities. Endocrine-related irAEs are more common in females, primarily driven by thyroid toxicity, which is typically mild. In contrast, immune-mediated hypophysitis, though less frequent, occurs more often in males and can lead to severe complications [62,63]. Furthermore, Zamami et al. reported a significant association between female sex and myocarditis, the most common immune-related cardiotoxicity [64].

In this context, Jing et al. conducted a large meta-analysis integrating pharmacovigilance data from FAERS and VigiBase with molecular omics data from The Cancer Genome Atlas (TCGA) to examine the association between sex and irAE risk. While no statistically significant difference was found (OR = 1.19; *p* = 0.21), female melanoma patients treated with anti-PD-1 and anti-PD-L1 therapies showed a higher tendency for irAEs [65]. However, these findings should be interpreted with caution, as data from pharmacovigilance databases and TCGA provided limited clinical details, such as menopausal status and comorbidities, which could significantly influence irAE risk. Additionally, since the total number of irAE cases served as a denominator in disproportionality analysis for signal detection, sex-biased reporting of adverse events occurrence and the higher incidence of cancers eligible for ICIs among males may influence results.

Recent studies have explored sex dimorphism in irAEs within patient populations with highly immunogenic tumor histologies, where sex-based immune differences may exacerbate irAE development. Duma et al. examined differences in the incidence of irAEs among pre-menopausal women, post-menopausal women, and men with metastatic melanoma (MM) and NSCLC treated with anti-PD-1 therapy. The study found that women with MM and NSCLC had a higher risk of irAEs than men when treated with anti-PD-1 inhibitors. Premenopausal women were more likely to develop irAEs compared with postmenopausal women and men (67% vs. 60% vs. 46%, *p* < 0.04). Notably, sex-based differences were also observed in the type of irAE, with women more commonly developing endocrinopathies, pneumonitis, and arthralgias [66]. Conversely, Bui et al. reported a higher incidence of dermatologic adverse events in female melanoma patients treated with ICIs compared to men, regardless of menopausal status. This suggests that certain irAEs may be more common in females due to mechanisms not necessarily linked to hormonal influence on immune response. External factors, such as skin type, environmental exposure, and pre-existing conditions, may contribute to the incidence of irAEs [67].

Among pre-existing factors, autoimmune predisposition may play a key role in irAE development. Notably, Cortellini et al. reported a higher incidence of irAEs in female patients and in those with pre-existing AIDs in a multivariate analysis. However, the higher prevalence of autoimmunity in women alone does not fully account for this finding [68].

Given the distinct mechanisms underlying immune activation, some studies have explored the association between specific immune biomarkers and irAE incidence from a sex-based perspective. Kudura et al. investigated whether higher immune activation in peripheral blood promotes the sex-biased onset of irAEs in patients with advanced melanoma treated with single (anti-PD-1) or dual-checkpoint inhibition (anti-PD-1/anti-CTLA-4). The study, which included 103 patients, found that irAEs were more frequent in women than in men (45% vs. 32%, *p* = 0.013), particularly endocrine irAEs (Table 1). Surprisingly, despite the higher incidence of irAEs, women exhibited lower immune activation in peripheral blood and a higher neutrophil-to-lymphocyte ratio (NLR) at baseline and six months after the initiation of ICI treatment [69]. Similarly, Wang et al. conducted a multivariate logistic analysis and identified female sex and a higher post-treatment NLR as factors correlating with an increased risk of developing irAEs in patients treated with anti-PD-1/PD-L1 inhibitors [70]. The lack of correlation between immune activation in the peripheral blood and irAE occurrence may reflect the complexity of immunological mechanisms beyond lymphocyte activation. Moreover, static time points may not adequately capture the dynamics of immune activation leading to specific toxicities during ICI treatment.

Valpione et al. evaluated 140 metastatic melanoma patients treated with Ipilimumab and found that women had a 1.5-fold increased risk of developing severe irAEs (Grade 3 or higher) compared to men (Table 1). Among all the biomarkers considered, female sex and lower IL-6 levels were independent risk factors for irAEs, suggesting distinct mechanisms in irAE occurrence within the context of the CTLA-4 pathway [71]. However, considering the wide age range of patients included in this study and the biphasic effect of estrogen on IL-6, it would be important to determine whether stratifying by menopausal status might change these results [72].

**Table 1 cancers-17-01054-t001:** A summary of studies that have reported the type and odds ratio of occurrence of immune-related adverse events (irAEs) in males and females.

Study	Pts	Type of Cancer	ICIs	irAE Type	Incidence (Female vs. Male)	OR (Female vs. Male) 95% CI [*p* = Value]
Unger et al. (2022) [59]	2319	Multiple cancer types	Anti-PD-1/PD-L1 +/− Anti-CTLA-4	Hematologic; Cardiotoxicity; Skin; Endocrine; GI; Neurological	56.6% vs. 48.8% (severe)	1.49 (1.24–1.78) *p* < 0.001
Duma et al. (2021) [66]	476	Melanoma; NSCLC	Anti-PD-1	Endocrine; Arthralgia; Pneumonitis	67% vs. 60% vs. 46%	1.12 (1.08–1.20) *p* < 0.04
Kudura et al. (2022) [69]	103	Melanoma	Anti-PD-1; Anti-PD-1 + Anti-CTLA-4	Skin; Thyroid,	45% vs. 32%	/
Bui et al. (2022) [67]	235	Melanoma	Anti-PD-1, PD-L1, CTLA4, PD-1 + CTLA-4	Dermatological	62.4% vs. 48.6%	2.1 (1.2–3.8) *p* = 0.01
Valpione et al. (2018) [71]	140	Melanoma	Anti-CTLA-4	Dermatological, GI, Endocrine	/	1.5 (1.06–2.16) *p* = 0.22
Jing et al. (2021) [65]	2982	Melanoma, NSCLC, RCC	Anti-PD-1, PD-L1	/	NSCLC 38.7% vs. 43.8%; Melanoma 39% vs. 45%	Pooled OR 1.19 (0.91–1.54); [*p* = 0.21] NSCLC: 0.97 (0.58–1.62); [*p* = 0.92] Melanoma 1.28 (1.01–1.63) [*p* = 0.04]
Miceli et al. (2023) [73]	204	Multiple cancer types	Anti-PD-1/PD-L1 +/− Anti-CTLA-4	Endocrine, hepatic	27% vs. 11%	/
Wahli et al. (2022) [74]	689	Multiple cancer types	Anti-PD-1, PD-L1, CTLA-4, Anti-PD-1 + Anti-CTLA-4		38.4% vs. 28.1%	/
Muir et al. (2021) [63]	1246	Melanoma	Anti-PD-1, Anti-PD-1 + Anti-CTLA-4	Thyroid	/	1.62 (1.27–2.08) *p* < 0.001

Anti-PD-1: anti-programmed death-1; PD-L1: programmed death-ligand 1; CTLA-4: cytotoxic T-lymphocyte-associated protein 4; NSCLC: non-small cell lung cancer.

Finally, sex dimorphism in the incidence and severity of irAEs may impact ICI treatment discontinuation and ultimately affect patient outcomes. In the ongoing prospective study by Miceli et al., which includes 204 patients, women have been found to be more likely to temporarily interrupt ICI treatment due to irAEs (27% vs. 11% in men) and experience a higher overall burden of irAEs across all grades [73].

Since the relationship between irAE occurrence and outcomes in patients treated with ICIs remains uncertain and given the potential variation in immunotherapy efficacy by sex, a different approach may be necessary in the future to optimize the balance between toxicity and efficacy from a sex-based perspective.

## 6. Mechanisms Underlying Sex Dimorphism in irAEs

Although recent evidence suggests a sex-based influence on the incidence and severity of irAEs in ICI-treated patients, the underlying causes remain largely unexplored.

Hormonal influences, autoimmunity, and tissue-specific differences may contribute to the severity of aberrant immune responses underlying the spectrum of irAEs. However, since some patients develop irAEs while others do not, it is possible that sex-based genetic differences such as specific HLA alleles and single nucleotide polymorphisms (SNPs) may play a key role in determining susceptibility to immunotherapy-related toxicities [75,76]. As seen with chemotherapy and targeted therapies, ICIs may also exhibit sex-specific pharmacokinetics due to inter-individual variability in monoclonal antibody disposition, mediated by both target-specific and non-specific receptor-mediated mechanisms [77,78,79,80].

Several population-based pharmacokinetic studies have shown that sex has a significant effect on the clearance of anti-PD-1/PD-L1 monoclonal antibodies (mAbs) across multiple cancer types, with clearance rates being lower in women compared to men [81].

The mechanisms underlying reduced clearance of ICIs in females are not entirely clear, but preclinical evidence suggests that sex hormones, particularly ER and PgR, may influence the degradation of ICIs through binding to Fcγ receptors (FcγRs) on various cells [80].

Specific target-mediated disposition may also play a role, due to sex-based differences in the expression of anti-PD-1/PD-L1 antibody ligands both within and outside the TME [82]. Mechanistically, these factors may lead to a higher relative dose and longer exposure time to ICIs in females, increasing the risk of irAE development.

Since ICIs modulate T-cell responses to antigens, sex hormones may enhance or suppress immune activation and irAE onset. For instance, the greater TCR repertoire clonality in females may increase extratumoral T-cell infiltration and cross-reactivity with self-antigens. Given that anti-PD-1/PD-L1 and CTLA-4 mAbs regulate different stages of T-cell development, sex-based differences in irAEs may vary depending on the type of ICI used [54]. Consistently, it has been demonstrated that women are at a higher risk of developing irAEs from anti-CTLA-4 treatment compared to men, likely due to dose-dependent mechanisms [83,84]. However, the same immune-related toxicity can be caused by both sex-dependent and sex-independent mechanisms, adding further complexity. Johnson et al. reported two cases of fulminant myocarditis associated with dual ICB (anti-PD1 and anti-CTLA-4), histologically characterized by robust T-cell and macrophage infiltration. Identical T-cell clonotypes were found in the heart, skeletal muscles, and tumor, suggesting shared antigenic targets as potential causes [85]. Additionally, preclinical studies in mice have shown that ICIs can cause myocardial damage by suppressing the transcription of cardioprotective factors such as MANF and HSPA5, which are normally positively regulated by E2 [86].

Beyond the role of T-cells, other adaptive immune pathways contribute to irAE development, including B cell activation and autoantibody production, mirroring mechanisms observed in AIDs. Self-tolerance in humans is partially maintained by the inhibition of auto-reactive T-cells through the CTLA-4 and PD-1/PD-L1 axes, and polymorphisms of PD-1 and CTLA-4 are associated with various autoimmune conditions [87,88]. Sex hormones regulate immune checkpoints, potentially contributing to irAE development. Estrogens, which play a protective role in kidney function, may help explain the lower incidence of immune-related nephrotoxicity observed in females by supporting PD-1/PD-L2-mediated self-tolerance mechanisms [61].

Additionally, evidence suggests that E2 may also increase the expression and function of PD-1 on T_reg_ cells, potentially modulating T_reg_ suppressive activity involved in autoimmunity [89]. In patients treated with anti-PD-1 who develop irAEs, an intense transcriptional reprogramming of T_reg_ cells has been demonstrated through mechanisms like those leading to autoimmune diseases [90]. In this context, E2 has been shown to modulate gene expression of Th17-mediated pathways and IL-17 production, which has been associated with severe irAEs, especially with anti-CTLA4 inhibitors [91,92,93]. Since the balance between Th17 and T_reg_ cells plays a role in the occurrence of irAEs, it is possible that ER contributes to dysregulated immune responses resulting from the disruption of the Th17/T_reg_ axis induced by ICB [94]. Due to increased T-cell activation during ICI treatment, enhanced T-cell/B-cell interactions can lead to autoantibody production, although their presence is not uniformly observed in all patients. The increased B-cell reactivity and autoantibody production in women contribute to a higher incidence of irAEs driven by these mechanisms. Furthermore, since anti-CTLA-4 promotes T-cell activation during the early immune response within lymphoid organs, it may increase the likelihood of the emergence of self-reactive clones and the severity of irAEs [91].

Different mechanisms underlying sex- and drug-specific irAEs are evident among endocrine toxicities, but whether an interaction exists remains unknown.

Immune-mediated thyroid dysfunction is more frequent in patients treated with anti-PD-1, particularly in women [63]. This may be attributed to pre-existing thyroid autoimmunity, characterized by a higher prevalence of thyroid-specific autoantibodies or thyroid-reactive T-cells, which are activated upon ICI treatment. Conversely, while primary hypophysitis is more common in women, immune-mediated hypophysitis occurs more frequently in men receiving anti-CTLA-4 therapy, suggesting distinct underlying pathogenic mechanisms (Figure 1) [55]. This rare but severe toxicity has been linked to a complement-mediated hypersensitivity reaction, driven by the expression of the CTLA-4 antigen on pituitary endocrine cells [95]. However, the underlying reasons for the male bias in the higher prevalence of this toxicity remain unclear, warranting further investigation.

Finally, emerging evidence suggests that environmental factors interact with the resident microbiome in organs like the skin and gastrointestinal tract, which are more exposed to external agents. Differences in gut microbiome composition may have a gender-based component, influenced not only by genetic and hormonal factors but also by dietary habits [96]. These variations contribute to distinct immune responses in males and females, potentially affecting both the risk and severity of irAEs. Women typically have greater gut microbial diversity with beneficial species like *Lactobacillus* and *Bifidobacterium*, which produce anti-inflammatory short-chain fatty acids (SCFAs) that support immune tolerance, potentially reducing severe irAEs. In contrast, men often have higher levels of *Bacteroides* species, which are associated with inflammatory responses and may heighten irAE susceptibility, particularly in the skin and gastrointestinal tract [97,98,99,100,101]. Female sex hormones further shape the gut microbiome composition through a bidirectional relationship. Estrogen levels correlate with intestinal microbiota diversity, and the anti-inflammatory effects of these hormones may differ between premenopausal and postmenopausal women. In postmenopausal women, lower estrogen levels are linked to reduced microbial diversity and increased inflammatory markers, reflecting a shift that may heighten the risk of irAEs [102].

## 7. Conclusions and Future Perspectives

The complex interplay between sex and the immune system contributes to significant differences in both the effectiveness of immunotherapy and the broad spectrum of associated immune-related toxicities.

This review underscores that sex-based differences can significantly influence the pharmacokinetics and immune responses associated with ICI treatment, leading to variations in the development of irAEs. Women generally experience a higher rate and distinct patterns of irAEs compared to men, primarily due to variations in immune regulatory pathways and hormonal influences that may interact differently with ICIs in each sex.

However, while the insights presented here highlight critical mechanisms underlying sex-based differences, they also reveal substantial gaps in clinical research. Much of the current understanding is based on preclinical models or retrospective studies, and evidence directly applicable to clinical practice remains limited. Moreover, past and current clinical trials often lack the necessary power to discern sex-based differences in irAE incidence and continue to reflect a greater representation of men compared to women.

Although a growing body of evidence suggests a trend toward a higher incidence of irAEs in females, conflicting results may be attributed to multiple contributing factors. The increased incidence of irAEs in the female sex may be associated with longer overall survival (OS), reflecting sex-based differences in survival outcomes with ICI therapy.

Additionally, significant heterogeneity across studies arises from the lack of systematic collection and analysis of sex-related key factors that influence immunotherapy outcomes. Evaluating the role of sex hormone regulation in different conditions is essential to understanding variations across immunotherapy strategies, tumor types, patient age, and menopausal status.

In the future, additional external host factors should be considered, including lifestyle (diet, smoking, and physical activity) and psychosocial influences (socioeconomic status, emotional well-being). This approach would support the transition from a “sex-based” to a more comprehensive “gender-based” perspective. Furthermore, gender may also affect treatment adherence, concomitant medications, and ultimately, the reported incidence of immune-related toxicities [103].

In this context, G-DEFINER is an ongoing observational, prospective, multicenter study designed to develop tools for individualized irAE risk prediction based on sex, gender, clinical, genetic, immunological, and microbiome factors. The study will conduct a multivariable analysis of differences that may contribute to varying irAE susceptibility between males and females, aiming to inform personalized immunotherapy approaches [104].

In conclusion, the evolving understanding of sex-related dimorphism in cancer treatment toxicities marks a pivotal shift toward more personalized and inclusive oncology. By advancing research in this direction, the field of oncology can improve the safety, efficacy, and equity of cancer care, ensuring that therapies are optimized for all patients, regardless of sex. This comprehensive approach promises to close existing knowledge gaps and achieve greater precision in cancer treatment, leading to better clinical outcomes across genders.

## Figures and Tables

**Figure 1 cancers-17-01054-f001:**
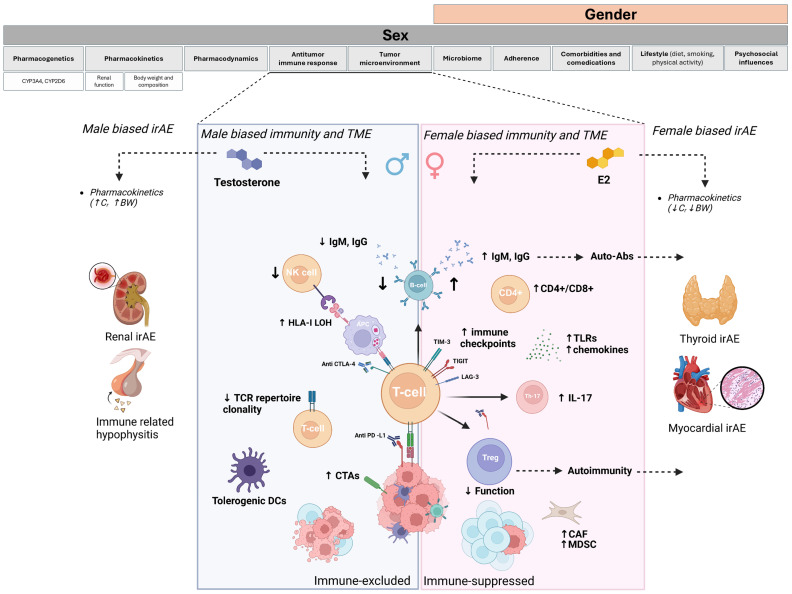
A graphical illustration of sex and gender = related differences in antitumor immune response and immune-related adverse events (irAE). CAF: cancer-associated fibroblast; CTAs: cancer–testis antigens; TCR: T-cell receptor; IgM: immunoglobulin M; IgG: immunoglobulin G; LAG3: lymphocyte-activation gene 3; TIGI T: T-cell immunoreceptor with Ig and ITIM domains; TIM-3: T-cell immunoglobulin and mucin-domain containing-3; IL-17: interleukin 17; HLA: human leukocyte antigen; DCs: dendritic cells.

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
