# Peer review of "Sex-Related Differences in Immunotherapy Toxicities: Insights into Dimorphic Responses"

_cancers, 2025, doi:10.3390/cancers17071054_

Round 1

Reviewer 1 Report

Comments and Suggestions for Authors

It is known that gender-related differences affect health status and impact on relevant outcomes in chronic pathologic conditions spanning from cardiovascular diseases to cancer. The manuscript “Sex-Related Differences in Immunotherapy Toxicities: Insights into Dimorphic Responses” is a review on sex-related differences in tumor therapy. The manuscript has the chapters:
- Biological Factors Underpinning Gender-Related Differences in Drug Toxicities: The Role of Sex Hormones and Pharmacokinetics.
- Sex differences in immunity and antitumor response
- Sex-Based Differences in Efficacy and Immune-Related Toxicities of Immune Checkpoint Inhibitors: Evidence and Insights.
The review is complemented by one Figure on “Graphical illustration of sex-related differences in the antitumour immune response and sex specific immune related adverse events (irAE)”, and one Table on “Summary of studies that have reported the type and odds ratio of occurrence of immune-related adverse events (irAEs) in males and females.”
At a glance, the manuscript seems well organized and written. 90 references have been included in the text, but a greater part of the references seems to be inaccurately assigned. Therefore, all references have to be revised!

  1. The authors have to better work out the differences to other reviews on the same theme
  2. The manuscript cites other reviews and not the original literature, e.g. reference [10] has been quoted 9 times and is a review on “Sex differences in immune responses”
  3. Some references do not fit the statement:
    -page 4, line 155-157: [15] has been cited for the statement “Thymic negative selection of autoreactive naïve T cells by autoimmune regulator (AIRE) gene is crucial for establishing central immune tolerance and has been shown to be sex hormone-dependent and less active in females”. However, the cited paper does not mention sex or hormone or female.
    -page 4, line 196-199: [22] has been cited for the statement “…the development of DCs is positively influenced by ER, enhancing both their quantity and functionality, which leads to increased T-cell trafficking, infiltration, and tumor cell recognition within the TME[22]”. However, the cited paper does not mention dendritic cells!
    -page 4, line199-202: [23] has been cited for the statement “In contrast, in males, DCs are polarized toward a tolerogenic phenotype due to androgen-mediated mechanisms, resulting in impaired neoantigen presentation. Additionally, men experience poorer immune infiltration, with a reduced T-cell receptor (TCR) repertoire clonality, lower expression levels of HLA class I and II molecules, and a higher frequency of HLA class I loss of heterozygosity (LOH) events”. However, the cited paper does mention dendritic cells 1 times, but not in context with male gender.
    -page 4, line 210-213: [26] has been cited for the statement “However, while T-cell activity may be impaired in both male and female TMEs, the mechanism differs. In females, TME is characterized by T-cell dysfunction despite high cytotoxic T lymphocyte (CTL), whereas TME in males shows T-cell exclusion with low CTL levels and a predominantly Th2-skewed response[26]”. However, the cited paper does not deal with gender differences or TME.
    -Page7, line326-331: [45] is taken as reference for the sentence “Conforti et al. showed that while higher TMB is generally associated with improved outcomes in patients treated with anti-PD-1/PD-L1 inhibitors, this relationship varies significantly between genders. While in males, the benefits of increased TMB are observed only at very high mutation levels, in females, there is a linear relationship between TMB and survival, suggesting a more efficient immune response and enhanced tumor recognition[45], [46].” However, [45]deals with immune-related adverse events!
    -Page 8 line 372 “Furthermore, Zamami et al. reported a significant association between female sex and myocarditis, the most common immune-related cardiotoxicity [45], [58]” Only [58] is Zamami et al, why the authors also cite [45]?
  4. Several statements lack references!
    a) e.g. line 128-129: “Estrogen and progesterone receptors re expressed in various cells of lymphoid tissue, including lymphocytes, macrophages and dendritic cells (DCs)”
    b) g. line 213-218: “To escape the early robust anti- tumor immune response in females, tumors may gradually develop immune evasion mechanisms. This leads to significantly higher expression of immunosuppressive cells, including macrophages, cancer-associated fibroblasts (CAFs), myeloid-derived suppressor cells (MDSCs), and Tregs, which characteristically express the X-linked FOXP3 gene and tend to be more activated in females.”
  5. Several references are incomplete (and could not/rarely been found), e.g.
    -ref. [11]: “Sohn, 2021; The importance of sex” which journal?
    -The year is lacking in [34];
    -journal is lacking in [8], [48], [46], [50], [69], [70], [71], [75], [88]
    -[63] R. Miceli et al., “2646 Poster Session Sex differences in burden of adverse events in patients receiving immunotherapy,” 727 2023. which journal?
    - Ref. [35] wrong journal: “Korean Association of Immunologists”, instead: Immune Netw. 2020 Feb 17;20(1):e9. doi: 10.4110/in.2020.20.e9
    -Ref [81] wrong journal: Chinese Soc Immunology?
  1. [22] = Ref. [27]
  2. Please explain “ER” in line 196-199 and line 284. “ER” cannot be found in the table of abbreviations. Similarly, RCTs should be explained in the table of abbreviations.
  3. Line 315-319: Could the authors better explain the sentence: “The study suggests that sex may influence the effectiveness of ICI treatments by impairing immunogenicity and neoantigen presentation through changes in the ER signaling pathway with aging[44].”
  4. Line 321-324: “Kugel et al found….[29]. Ref. 29 is Castro et al
  5. The authors have to refer to Fig. 1 in the context of the whole manuscript text.

Author Response

Dear Reviewer 1,

We sincerely thank you for accepting the assignment and for your thorough review of the bibliography. The errors in the bibliography were due to an issue with the reference manager and have now been corrected, along with the addition of the missing references. Furthermore, whenever possible, references to literature reviews have been replaced with original studies. Additionally, Figure 1 has been further integrated into the context of the paper, and the relevant link has been included where appropriate.

Best regards,

The Authors

Reviewer 2 Report

Comments and Suggestions for Authors

This manuscript provides a comprehensive review of sex-related differences in immune checkpoint inhibitor (ICI) therapies, a topic that many may not be fully aware of. The authors offer an excellent summary of the field, covering the role of sex hormones and pharmacokinetics, as well as sex differences in immunity and antitumor responses. After giving this introductive information, they discuss sex-based differences in the efficacy and immune-related toxicities associated with immune checkpoint inhibitors. The accompanying figure effectively illustrates these concepts, and the table provides useful information for readers. The review is also well-referenced.

Author Response

Dear Reviewer 2,

We sincerely thank you for accepting the review and for your positive feedback. We remain available for any further revisions or suggestions to enhance the quality of the manuscript.

Best regards,
The Authors.

Reviewer 3 Report

Comments and Suggestions for Authors

The article "Sex-Related Differences in Immunotherapy Toxicities: Insights into Dimorphic Responses" analyzes how biological sex affects immune responses and the efficacy of immunotherapy in cancer treatment. It highlights sex-based disparities in pharmacokinetics, pharmacodynamics, and immune-related adverse events (irAEs) associated with immune checkpoint inhibitors (ICIs). The paper stresses the need for sex-specific strategies in oncology to improve therapeutic efficacy and safety by integrating preclinical studies, clinical research, and pharmacovigilance data. This is an insightful study, and addressing the limitations outlined could further enhance the impact and applicability of its findings. The review places disproportionate emphasis on preclinical data, highlighting the need for more human studies to improve translational relevance. Many cited studies rely on retrospective analyses that lack essential clinical details (e.g., menopausal status, comorbidities), with prospective trials like G-DEFINER being infrequent. Additionally, conflicting clinical findings, such as those between Wallis et al. and Conforti et al. on ICI efficacy, are acknowledged but not critically examined, which could undermine the review’s conclusions.

A few suggestions to further strengthen your manuscript:

  1. How do you tackle the possible complications that emerge from the inconsistent differentiation between “sex” (biological characteristics) and “gender” (social and cultural influences) in your analysis?
    Could you elaborate on your literature selection technique to guarantee that the evaluation is both thorough and replicable?
    3. Considering that numerous mechanistic breakthroughs originate from preclinical models, what is your perspective on their direct relevance to human clinical situations?
    4. What measures have you implemented—or could be implemented—to systematically mitigate significant confounding variables such as menopausal status, age-related immunosenescence, and comorbidities?
    5. Do you believe that the extensive scope of the review may have undermined the thoroughness of the investigation about the molecular pathways that account for variations in medication toxicity?
    6. Can you discuss the possible effects of disparate adverse event reporting between genders, which may skew the reported incidence of immune-related adverse events?
    7. Could unmeasured or uncontrolled confounders, such as lifestyle factors, comorbidities, or socioeconomic status, be affecting the observed sex disparities in immunotherapy responses?
    8. How are variations in treatment regimens—such as monotherapy compared to combination therapy—considered when evaluating sex-based discrepancies in efficacy and toxicity?
    9. Might the aggregation of data from many cancer types obscure potential disease-specific impacts on the sex-related disparities noted in immune checkpoint inhibitor responses?

Although this study provides important insights into  sex-based disparities in pharmacokinetics, pharmacodynamics, and immune-related adverse events (irAEs) associated with immune checkpoint inhibitors (ICIs) and other, I believe further improvements and revisions are needed before publication

Author Response

The authors would like to sincerely thank Reviewer 3 for the valuable insights and thoughtful feedback. We have addressed the comments to the best extent possible while ensuring consistency with the objectives of the manuscript.

We remain available for any further suggestions or improvements.

Best regards,

the authors.

Response to Reviewer 3 comments:

Comment 1. How do you tackle the possible complications that emerge from the inconsistent differentiation between “sex” (biological characteristics) and “gender” (social and cultural influences) in your analysis?
Could you elaborate on your literature selection technique to guarantee that the evaluation is both thorough and replicable?

We thank the reviewer for pointing this issue out and allowing us to better clarify this point.

A significant portion of the existing literature on sex-based differences in immunotherapy efficacy and toxicity focuses on biological differences, as extensively discussed in the manuscript, including genetic and hormonal variations. The broader perspective of assessing “gender” differences is a highly recent topic with limited data available in the literature. However, this important aspect has been incorporated into Figure 1 and the manuscript's conclusions as a crucial area for future research to ensure a more comprehensive evaluation. Additionally, as suggested, the methodology used for the literature review underlying this manuscript has been included.

Comment 2. Considering that numerous mechanistic breakthroughs originate from preclinical models, what is your perspective on their direct relevance to human clinical situations?

Comment 3. What measures have you implemented—or could be implemented—to systematically mitigate significant confounding variables such as menopausal status, age-related immunosenescence, and comorbidities?

We thank the reviewer for raising this point. A fundamental limitation of the current data on the mechanisms underlying sex differences in immunotherapy is that they are largely derived from preclinical models. We have highlighted this important bias in the manuscript and consider it a critical issue that should be addressed through prospective studies incorporating biomarker collection and analysis during immunotherapy, as proposed in the G-DEFINER clinical trial.

Furthermore, stratifying patients based on key factors such as BMI, gender (which encompasses comorbidities, co-medications, and treatment adherence), hormonal status, and age—potentially using indirect biomarkers of immunosenescence—is essential for prospectively assessing differences in treatment outcomes and toxicity.

Comment 4. Do you believe that the extensive scope of the review may have undermined the thoroughness of the investigation about the molecular pathways that account for variations in medication toxicity?

We thank the reviewer for the insightful suggestion. In the chapter dedicated to potential mechanisms influencing immunotherapy toxicity from a sex-based perspective, we aimed to present the pathways and alterations with the strongest evidence in the literature. Additionally, we highlighted specific cases of rare but highly severe toxicities, such as immune-related myocarditis and hypophysitis, along with their underlying biological and molecular mechanisms, as these conditions exhibit the most substantial evidence of sex-specific differences.

Comment 5. Can you discuss the possible effects of disparate adverse event reporting between genders, which may skew the reported incidence of immune-related adverse events?

Thank you for this important question. This type of assessment is the first factor that must be addressed when analyzing sex differences. As mentioned in the manuscript, this factor is inherently gender-related, leading to a tendency for higher reporting of adverse events in females compared to males. However, for many of the immune-related toxicities discussed in the manuscript (as well as in the study by Unger et al.), the higher incidence in females was observed not only for symptomatic but also for objective adverse events. This underscores a potentially real difference that extends beyond gender-related reporting bias.

Comment 6. Could unmeasured or uncontrolled confounders, such as lifestyle factors, comorbidities, or socioeconomic status, be affecting the observed sex disparities in immunotherapy responses?

It is highly plausible that gender-based factors, such as lifestyle, comorbidities, and socioeconomic status, influence immunotherapy efficacy and toxicity. However, no robust data in the literature currently confirm this association, as these factors are not typically included in multivariate analyses of prospective or retrospective clinical studies. We appreciate this valuable opportunity to further explore this aspect, as the manuscript aims to emphasize the need for deeper investigation into both biological sex differences and the broader gender-related factors in the future of clinical research.

Comment 7. How are variations in treatment regimens—such as monotherapy compared to combination therapy—considered when evaluating sex-based discrepancies in efficacy and toxicity?

Thank you for this insightful reflection, which provides an opportunity to further explore this aspect. In the manuscript, we discussed potential sex-specific differences not only in the efficacy but also in the toxicity of immune checkpoint inhibitors, both as monotherapy and in combination with dual checkpoint blockade. Additionally, we highlighted the immunological mechanisms that may underlie the differences in type and severity of toxicities observed with dual immune blockade. However, an interesting topic not covered in the paper—since our focus was specifically on immunotherapy—is whether the combination with chemotherapy (and its associated hormonal effects) or other type of drugs may influence sex-based variations in immune-related adverse events (irAE). This remains an area of ongoing research, and we are currently reviewing the available literature on this subject.

Comment 8. Might the aggregation of data from many cancer types obscure potential disease-specific impacts on the sex-related disparities noted in immune checkpoint inhibitor responses?

Thank you for this highly relevant comment. In our view, the heterogeneous findings from studies assessing sex-specific differences in both efficacy and incidence of immune-related adverse events (irAE) are largely influenced by the inclusion of tumor types with inherent sex-related biases. Tumor histology may impact immunogenicity, tumor microenvironment (TME) characteristics, and specific tissue targets, potentially driving cross-reactivity—one of the key mechanisms underlying irAE development—from a sex-based perspective. Future research should focus on systematic retrospective analyses that account for these factors, along with the specific type of immunotherapy used and the sex-related variables outlined in the manuscript. Such an approach would help generate more robust and less heterogeneous data compared to what is currently available in the literature.

Reviewer 4 Report

Comments and Suggestions for Authors

Canzian and colleagues provide a review article focused on gender dimorphism in benefits vs. irAEs associated with immunotherapeutic intervention in cancer patients. The topic is timely and underscores the need to integrate sex as an important clinical variable in developing safe and effective immunotherapy programs for patients with cancer. Important issues including gender-associated differences in drug pharmacodynamics/pharmacokinetics, immunology/TIME composition/function, sex hormones/age, severity and organ/system targeting of irAEs, and proposed mechanisms of action including the influence of the patient’s skin/gut microbiome. While many of the references appear incomplete (requiring reinspection and correction), they are current, topically comprehensive and unbiased in their selection. Future directions for expanding and extracting actionable data from gender-based studies (including clinical trials) is foreshadowed. The review is well-written and will serve as a useful primer to those new to this area of cancer research and an effective update to the more experienced reader. In addition to the request for editing of the references, my only other comment related to areas for improvement would be current lack of discussing the substantial published literature relevant to inhibition/antagonism of sex steroids/receptors as these relate to cancer patient response to immunotherapy/irAEs. Inclusion of such information as a separate chapter and its integration in the Conclusion/Future Perspectives section would improve the report’s overall merit.  

Author Response

Dear Reviewer 2,

We sincerely thank you for accepting the review and for your positive feedback. 

We fully agree that understanding how the pharmacological modulation of sex hormone pathways influences the efficacy and safety of immunotherapy is a topic of great future interest. However, as this is an emerging area with limited available data in the literature, it has not been included in the manuscript to maintain focus on sex differences in immune-related toxicities.

We remain available for any further revisions or suggestions to enhance the quality of the manuscript.

Best regards, 

The Authors

Reviewer 5 Report

Comments and Suggestions for Authors

the paper explains an important and underexplored field of cancer immunotherapy, with a focus on the effect of biological sex on treatment. it also moves from setting the background to stating the problem and its implications easily, making it easy to read. however, I have some concerns:

  • Refine phrasing for clarity and precision throughout the manuscript. example: sex dimorphism in the safety and effectiveness of cancer immunotherapy remains an underexplored area in the abstract doesn't sound good. I do see this in other sections.
  • I suggest including methodology in this review. paper from which years to years, refine search words. 
  • The manuscript discusses studies but lacks citation to enough actual studies. More fact-supported evidence (i.e., numerical data for sex-based toxicological difference) would support the claims more authoritatively.

  •  Some points of argumentation (e.g., underrepresentation of women in clinical trials) repeat each other using different phrasing. Clarification by omission of redundant text could make reading more effective.

  • chapter 4 identifies differences in outcomes but does not provide clear solutions to minimize study heterogeneity (e.g., standardizing patient populations, biomarkers).
  •  Mentions that chemotherapy may enhance ICI response in women but does not mention other potential combination strategies (e.g., targeted therapies, hormonal modulation).
  • Figure 1 is referred to but not described in detail and its significance is not explained.
  •  Some topics, such as the influence of sex on immune response and tumor immune evasion, are discussed again under different sections, making the discussion a bit repetitive.
  • The sentences in some sections are occasionally extremely long and complex. It would be preferable to have them broken into smaller ones.
  • The subject jumps around between discussions (e.g., pharmacokinetics, T-cell responses, microbiome) with no bridging sentences to create logical flow.
  • please add few more pictures and provide a more explicit mechanistic explanation of sex differences in immune responses.
  • please add limitation of this review article 

Author Response

Dear Reviewer,

We sincerely appreciate your valuable suggestions for improving the structural and formal aspects of the paper. We have conducted a major revision, enhancing fluency, logical coherence, and removing redundant or repetitive concepts. Additionally, we have included the literature search methodology underlying this manuscript.

Regarding specific revision requests, we note the following:

  1. The manuscript discusses studies but lacks citation to enough actual studies. More fact-supported evidence (i.e., numerical data for sex-based toxicological difference) would support the claims more authoritatively.
  1. The cited studies represent the most recent and well-supported evidence available in the literature on this largely unexplored topic. For clarity and text fluidity, we have incorporated the toxicity data from the referenced studies into Table 1 to facilitate readability and improve the manuscript’s structure.
  1. Chapter 4 identifies differences in outcomes but does not provide clear solutions to minimize study heterogeneity (e.g., standardizing patient populations, biomarkers).
    1. This aspect was not addressed in detail, as the focus of the review is on reported toxicities based on the available literature rather than an extensive evaluation of the reasons behind differing efficacy outcomes and potential solutions. However, we have incorporated a critical assessment of the possible causes underlying the heterogeneity of studies evaluating sex differences in immunotherapy efficacy.
  2. Mentions that chemotherapy may enhance ICI response in women but does not mention other potential combination strategies (e.g., targeted therapies, hormonal modulation).
    1. For the same reason, further differences in immunotherapy efficacy with other treatment combinations beyond chemotherapy were not explored in depth, as this falls outside the scope of the present manuscript, despite its recognized relevance.
  1. Please add limitation of this review article 
    1. We have incorporated what we consider to be the intrinsic limitations of this review, now reflected in the conclusions. The primary limitation remains the nature of the currently available data, which are often derived from heterogeneous cohorts and lack the collection of critical sex-based factors. “However, while While the insights presented here highlight critical mechanisms underlying sex-based differences, they also reveal substantial gaps in clinical research. Much of the current understanding is based on animal preclinical models or retrospective studies, and evidence directly applicable to clinical practice remains limited. Moreover, past and current clinical trials often lack the necessary power to discern sex-based differences in irAE incidence and continue to reflect a greater representation of men compared to women.” “Additionally, significant heterogeneity across studies arises from the lack of systematic collection and analysis of sex-based key factors influencing immunotherapy outcomes. Evaluating the role of sex hormones regulation in different conditions is essential to understanding variations across immunotherapy strategies, tumor types, patient age, and menopausal status.”

Round 2

Reviewer 3 Report

Comments and Suggestions for Authors

Accept

Comments on the Quality of English Language

None

Reviewer 5 Report

Comments and Suggestions for Authors

Thank you for revising the paper. All the best.